# Training Image Estimators
# without Image Ground-Truth

**Zhihao Xia**
Washington University in St. Louis
1 Brookings Dr., St. Louis, MO 63130
zhihao.xia@wustl.edu

**Ayan Chakrabarti**
Washington University in St. Louis
1 Brookings Dr., St. Louis, MO 63130
ayan@wustl.edu

## Abstract

Deep neural networks have been very successful in image estimation applications such as compressive-sensing and image restoration, as a means to estimate images from partial, blurry, or otherwise degraded measurements. These networks are trained on a large number of corresponding pairs of measurements and ground-truth images, and thus implicitly learn to exploit domain-specific image statistics. But unlike measurement data, it is often expensive or impractical to collect a large training set of ground-truth images in many application settings. In this paper, we introduce an unsupervised framework for training image estimation networks, from a training set that contains only measurements—with two varied measurements per image—but no ground-truth for the full images desired as output. We demonstrate that our framework can be applied for both regular and blind image estimation tasks, where in the latter case parameters of the measurement model (e.g., the blur kernel) are unknown: during inference, and potentially, also during training. We evaluate our method for training networks for compressive-sensing and blind deconvolution, considering both non-blind and blind training for the latter. Our unsupervised framework yields models that are nearly as accurate as those from fully supervised training, despite not having access to any ground-truth images.

## 1   Introduction

Reconstructing images from imperfect observations is a classic inference task in many imaging applications. In compressive sensing [8], a sensor makes partial measurements for efficient acquisition. These measurements correspond to a low-dimensional projection of the higher-dimensional image signal, and the system relies on computational inference for recovering the full-dimensional image. In other cases, cameras capture degraded images that are low-resolution, blurry, etc., and require a restoration algorithm [10, 29, 34] to recover a corresponding un-corrupted image. Deep convolutional neural networks (CNNs) have recently emerged as an effective tool for such image estimation tasks [4, 6, 7, 12, 27, 30, 31]. Specifically, a CNN for a given application is trained on a large dataset that consists of pairs of ground-truth images and observed measurements (in many cases where the measurement or degradation process is well characterized, having a set of ground-truth images is sufficient to generate corresponding measurements). This training set allows the CNN to learn to exploit the expected statistical properties of images in that application domain, to solve what is essentially an ill-posed inverse problem.

But for many domains, it is impractical or prohibitively expensive to capture full-dimensional or un-corrupted images, and construct such a large representative training set. Unfortunately, it is often in such domains that a computational imaging solution is most useful. Recently, Lehtinen *et al*. [14] proposed a solution to this issue for denoising, with a method that trains with only pairs of noisy observations. While their method yields remarkably accurate network models without needing any

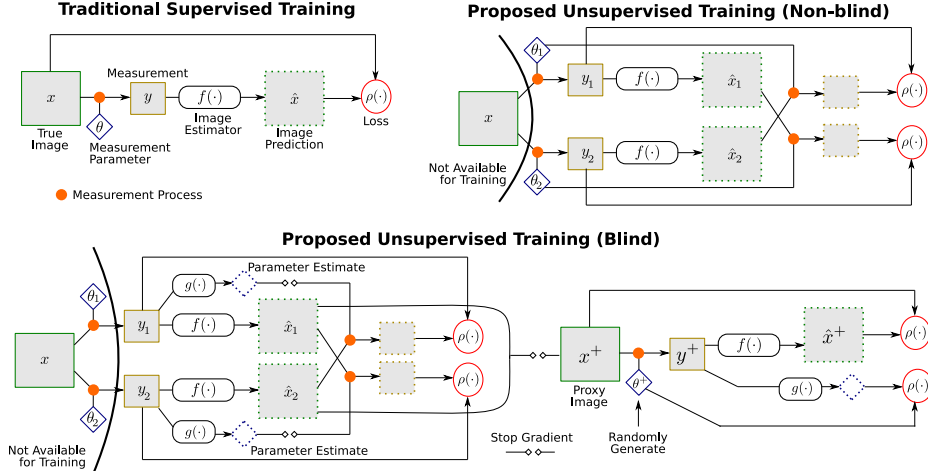

Figure 1: **Unsupervised Training from Measurements.** Our method allows training image estimation networks $f(\cdot)$ from sets of pairs of varied measurements, but without the underlying ground-truth images. *(Top Right)* We supervise training by requiring that network predictions from one measurement be consistent with the other, when measured with the corresponding parameter. *(Bottom)* In the blind training setting, when both the image and measurement parameters are unavailable, we also train a parameter estimator $g(\cdot)$. Here, we generate a proxy training set from the predictions of the model (as it is training), and use synthetic measurements from these proxies to supervise training of the parameter estimator $g(\cdot)$, and augment training of the image estimator $f(\cdot)$.

ground-truth images for training, it is applicable only to the specific case of estimation from noisy measurements—when each image intensity is observed as a sample from a (potentially unknown) distribution with mean or mode equal to its corresponding true value.

In this work, we introduce an unsupervised method for training image estimation networks that can be applied to a general class of observation models—where measurements are a linear function of the true image, potentially with additive noise. As training data, it only requires two observations for the same image but not the underlying image itself[1]. The two measurements in each pair are made with different parameters (such as different compressive measurement matrices or different blur kernels), and these parameters vary across different pairs. Collecting such a training set provides a practical alternative to the more laborious one of collecting full image ground-truth. Given these measurements, our method trains an image estimation network by requiring that its prediction from one measurement of a pair be consistent with the other measurement, when observed with the corresponding parameter. With sufficient diversity in measurement parameters for different training pairs, we show this is sufficient to train an accurate network model despite lacking direct ground-truth supervision.

While our method requires knowledge of the measurement *model* (e.g., blur by convolution), it also incorporates a novel mechanism to handle the blind setting during training—when the measurement *parameters* (e.g., the blur kernels) for training observations are unknown. To be able to enforce consistency as above, we use an estimator for measurement parameters that is trained simultaneously using a "proxy" training set. This set is created on-the-fly by taking predictions from the image network even as it trains, and pairing them with observations synthetically created using randomly sampled, and thus known, parameters. The proxy set provides supervision for training the parameter estimator, and to augment training of the image estimator as well. This mechanism allows our method to nearly match the accuracy of fully supervised training on image and parameter ground-truth.

We validate our method with experiments on image reconstruction from compressive measurements and on blind deblurring of face images, with blind and non-blind training for the latter, and compare to fully-supervised baselines with state-of-the-art performance. The supervised baselines use a training set of ground-truth images and generate observations with random parameters on the fly in each epoch, to create a much larger number of effective image-measurement pairs. In contrast, our method is trained with only two measurements per image from the same training set (but not the image

itself), with the pairs kept fixed through all epochs of training. Despite this, our unsupervised training method yields models with test accuracy close to that of the supervised baselines, and thus presents a practical way to train CNNs for image estimation when lacking access to image ground truth.

## 2  Related Work

**CNN-based Image Estimation.** Many imaging tasks require inverting the measurement process to obtain a clean image from the partial or degraded observations—denoising [3], deblurring [29], super-resolution [10], compressive sensing [8], etc. While traditionally solved using statistical image priors [9, 25, 34], CNN-based estimators have been successfully employed for many of these tasks. Most methods [4, 6, 7, 12, 22, 27, 30, 31] learn a network to map measurements to corresponding images from a large training set of pairs of measurements and ideal ground-truth images. Some learn CNN-based image priors, as denoisers [5, 23, 31] or GANs [1], that are agnostic to the inference task (denoising, deblurring, etc.), but still tailored to a chosen class of images. All these methods require access to a large domain-specific dataset of ground-truth images for training. However, capturing image ground-truth is burdensome or simply infeasible in many settings (e.g., for MRI scans [18] and other biomedical imaging applications). In such settings, our method provides a practical alternative by allowing estimation networks to be trained from measurement data alone.

**Unsupervised Learning.** Unsupervised learning for CNNs is broadly useful in many applications where large-scale training data is hard to collect. Accordingly, researchers have proposed unsupervised and weakly-supervised methods for such applications, such as depth estimation [11, 32], intrinsic image decomposition [16, 19], etc. However, these methods are closely tied to their specific applications. In this work, we seek to enable unsupervised learning for image estimation networks. In the context of image modeling, Bora *et al.* [2] propose a method to learn a GAN model from only degraded observations. Their method, like ours, includes a measurement model with its discriminator for training (but requires knowledge of measurement parameters, while we are able to handle the blind setting). Their method proves successful in training a generator for ideal images. We seek a similar unsupervised means for training image reconstruction and restoration networks.

The closest work to ours is the recent *Noise2Noise* method of Lehtinen *et al.* [14], who propose an unsupervised framework for training denoising networks by training on pairs of noisy observations of the same image. In their case, supervision comes from requiring the denoised output from one observation be close to the other. This works surprisingly well, but is based on the assumption that the expected or median value of the noisy observations is the image itself. We focus on a more general class of observation models, which requires injecting the measurement process in loss computation. We also introduce a proxy training approach to handle blind image estimation applications.

Also related are the works of Metzler *et al.* [21] and Zhussip *et al.* [33], that use Stein's unbiased risk estimator for unsupervised training from only measurement data, for applications in compressive sensing. However, these methods are specific to estimators based on D-AMP estimation [20], since they essentially train denoiser networks for use in unrolled AMP iterations for recovery from compressive measurements. In contrast, ours is a more general framework that can be used to train generic neural network estimators.

## 3  Proposed Approach

Given a measurement $y \in \mathbb{R}^M$ of an ideal image $x \in \mathbb{R}^N$ that are related as

$$y = \theta\, x + \epsilon, \tag{1}$$

our goal is to train a CNN to produce an estimate $\hat{x}$ of the image from $y$. Here, $\epsilon \sim p_\epsilon$ is random noise with distribution $p_\epsilon(\cdot)$ that is assumed to be zero-mean and independent of the image $x$, and the parameter $\theta$ is an $M \times N$ matrix that models the linear measurement operation. Often, the measurement matrix $\theta$ is structured with fewer than $MN$ degrees of freedom based on the measurement model—e.g., it is block-Toeplitz for deblurring with entries defined by the blur kernel. We consider both non-blind estimation when the measurement parameter $\theta$ is known for a given measurement during inference, and the blind setting where $\theta$ is unavailable but we know the distribution $p_\theta(\cdot)$. For blind estimators, we address both non-blind and blind training—when $\theta$ is known for each measurement in the training set but not at test time, and when it is unknown during training as well.

Since (1) is typically non-invertible, image estimation requires reasoning with the statistical distribution $p_x(\cdot)$ of images for the application domain, and conventionally, this is provided by a large training set of typical ground-truth images $x$. In particular, CNN-based image estimation methods train a network $f : y \to \hat{x}$ on a large training set $\{(x_t, y_t)\}_{t=1}^{T}$ of pairs of corresponding images and measurements, based on a loss that measures error $\rho(\hat{x}_t - x_t)$ between predicted and true images across the training set. In the non-blind setting, the measurement parameter $\theta$ is known and provided as input to the network $f$ (we omit this in the notation for convenience), while in the blind setting, the network must also reason about the unknown measurement parameter $\theta$.

To avoid the need for a large number of ground-truth training images, we propose an unsupervised learning method that is able to train an image estimation network using measurements alone. Specifically, we assume we are given a training set of two measurements $(y_{t:1}, y_{t:2})$ for each image $x_t$:

$$y_{t:1} = \theta_{t:1}\, x_t + \epsilon_{t:1}, \quad y_{t:2} = \theta_{t:2}\, x_t + \epsilon_{t:2}, \tag{2}$$

but not the images $\{x_t\}$ themselves. We require the corresponding measurement parameters $\theta_{t:1}$ and $\theta_{t:2}$ to be different for each pair, and further, to also vary across different training pairs. These parameters are assumed to be known for the non-blind training setting, but not for blind training.

## 3.1 Unsupervised Training for Non-Blind Image Estimation

We begin with the simpler case of non-blind estimation, when the parameter $\theta$ for a given measurement $y$ is known, both during inference and training. Given pairs of measurements with known parameters, our method trains the network $f(\cdot)$ using a "swap-measurement" loss on each pair, defined as:

$$\mathcal{L}_{\text{swap}} = \frac{1}{T} \sum_t \rho\Big(\theta_{t:2}\, f(y_{t:1}) \;-\; y_{t:2}\Big) + \rho\Big(\theta_{t:1}\, f(y_{t:2}) \;-\; y_{t:1}\Big). \tag{3}$$

This loss evaluates the accuracy of the full images predicted by the network from each measurement in a pair, by comparing it to the other measurement—using an error function $\rho(\cdot)$—after simulating observation with the corresponding measurement parameter. Note *Noise2Noise* [14] can be seen as a special case of (3) for measurements are degraded only by noise, with $\theta_{t:1} = \theta_{t:2} = I$.

When the parameters $\theta_{t:1}, \theta_{t:2}$ used to acquire the training set are sufficiently diverse and statistically independent for each underlying $x_t$, this loss provides sufficient supervision to train the network $f(\cdot)$. To see this, we consider using the $L_2$ distance for the error function $\rho(z) = \|z\|^2$, and note that (3) represents an empirical approximation of the expected loss over image, parameter, and noise distributions. Assuming the training measurement pairs are obtained using (2) with $x_t \sim p_x$, $\theta_{t:1}, \theta_{t:2} \sim p_\theta$, and $\epsilon_{t:1}, \epsilon_{t:2} \sim p_\epsilon$ drawn i.i.d. from their respective distributions, we have

$$\mathcal{L}_{\text{swap}} \approx 2 \mathop{\mathbb{E}}_{x \sim p_x} \mathop{\mathbb{E}}_{\theta_1 \sim p_\theta} \mathop{\mathbb{E}}_{\epsilon_1 \sim p_\epsilon} \mathop{\mathbb{E}}_{\theta_2 \sim p_\theta} \mathop{\mathbb{E}}_{\epsilon_2 \sim p_\epsilon} \|\theta_2 f(\theta_1 x + \epsilon_1) - (\theta_2 x + \epsilon_2)\|^2$$

$$= 2\sigma_\epsilon^2 + 2 \mathop{\mathbb{E}}_{x \sim p_x} \mathop{\mathbb{E}}_{\theta \sim p_\theta} \mathop{\mathbb{E}}_{\epsilon \sim p_\epsilon} \Big(f(\theta x + \epsilon) \;-\; x\Big)^T Q \Big(f(\theta x + \epsilon) \;-\; x\Big), \quad Q = \mathop{\mathbb{E}}_{\theta' \sim p_\theta}(\theta'^T \theta'). \tag{4}$$

Therefore, because the measurement matrices are independent, we find that in expectation the swap-measurement loss is equivalent to supervised training against the true image $x$, with an $L_2$ loss that is weighted by the $N \times N$ matrix $Q$ (upto an additive constant given by noise variance). When the matrix $Q$ is full-rank, the swap-measurement loss will provide supervision along all image dimensions, and will reach its theoretical minimum $(2\sigma_\epsilon^2)$ *iff* the network makes exact predictions.

The requirement that $Q$ be full-rank implies that the distribution $p_\theta$ of measurement parameters must be sufficiently diverse, such that the full *set* of parameters $\{\theta\}$, used for training measurements, together span the entire domain $\mathbb{R}^N$ of full images. Therefore, even though measurements made by individual $\theta$—and even pairs of $(\theta_{t:1}, \theta_{t:2})$—are incomplete, our method relies on the fact that the full set of measurement parameters used during training *is* complete. Indeed, for $Q$ to be full-rank, it is important that there be no systematic deficiency in $p_\theta$ (e.g., no vector direction in $\mathbb{R}^N$ left unobserved by all measurement parameters used in training). Also note that while we derived (4) for the L2 loss, the argument applies to any error function $\rho(\cdot)$ that is minimized only when its input is 0.

In addition to the swap loss, we also find it useful to train with an additional "self-measurement" loss that measures consistency between an image prediction and its own corresponding input measurement:

$$\mathcal{L}_{\text{self}} = \frac{1}{T} \sum_t \rho\Big(\theta_{t:1}\, f(y_{t:1}) \;-\; y_{t:1}\Big) + \rho\Big(\theta_{t:2}\, f(y_{t:2}) \;-\; y_{t:2}\Big). \tag{5}$$

While not sufficient by itself, we find the additional supervision it provides to be practically useful in yielding more accurate network models since it provides more direct supervision for each training sample. Therefore, our overall unsupervised training objective is a weighted version of the two losses $\mathcal{L}_{\text{swap}} + \gamma\mathcal{L}_{\text{self}}$, with weight $\gamma$ chosen on a validation set.

## 3.2 Unsupervised Training for Blind Image Estimation

We next consider the more challenging case of blind estimation, when the measurement parameter $\theta$ for an observation $y$ is unknown—and specifically, the blind training setting, when it is unknown even during training. The blind training setting complicates the use of our unsupervised losses in (3) and (5), since the values of $\theta_{t:1}$ and $\theta_{t:2}$ used there are unknown. Also, blind estimation tasks often have a more diverse set of possible parameters $\theta$. While supervised training methods with access to ground-truth images can generate a very large database of synthetic image-measurement pairs by pairing the same image with many different $\theta$ (assuming $p_\theta(\cdot)$ is known), our unsupervised framework has access only to two measurements per image.

However, in many blind estimation applications (such as deblurring), the parameter $\theta$ has comparatively limited degrees of freedom and the distribution $p_\theta(\cdot)$ is known. Consequently, it is feasible to train estimators for $\theta$ from an observation $y$ with sufficient supervision. With these assumptions, we propose a "proxy training" approach for unsupervised training of blind image estimators. This approach treats estimates from our network during training as a source of image ground-truth to train an estimator $g : y \to \hat{\theta}$ for measurement parameters. We use the image network's predictions to construct synthetic observations as:

$$x_{t:i}^+ \leftarrow f(y_{t:i}), \;\; \theta_{t:i}^+ \sim p_\theta, \;\; \epsilon_{t:i}^+ \sim p_\epsilon, \quad y_{t:i}^+ = \theta_{t:i}^+ \, x_{t:i}^+ + \epsilon_{t:i}^+, \quad \text{for } i \in \{1,2\}, \tag{6}$$

where $\theta_{t:i}^+$ and $\epsilon_{t:i}^+$ are sampled on the fly from the parameter and noise distributions, and $\leftarrow$ indicates an assignment with a "stop-gradient" operation (to prevent loss gradients on the proxy images from affecting the image estimator $f(\cdot)$). We use these synthetic observations $y_{t:i}^+$, with known sampled parameters $\theta_{t:i}^+$, to train the parameter estimation network $g(\cdot)$ based on the loss:

$$\mathcal{L}_{\text{prox}:\theta} = \frac{1}{T} \sum_t \sum_{i=1}^2 \rho\left(g(y_{t:i}^+) \;-\; \theta_{t:i}^+\right). \tag{7}$$

As the parameter network $g(\cdot)$ trains with augmented data, we simultaneously use it to compute estimates of parameters for the original observations: $\hat{\theta}_{t:i} \leftarrow g(y_{t:i}), \quad \text{for } i \in \{1,2\}$, and compute the swap- and self-measurement losses in (3) and (5) on the original observations using these estimated, instead of true, parameters. Notice that we use a stop-gradient here as well, since we do not wish to train the parameter estimator $g(\cdot)$ based on the swap- or self-measurement losses—the behavior observed in (4) no longer holds in this case, and we empirically observe that removing the stop-gradient leads to instability and often causes training to fail.

In addition to training the parameter estimator $g(\cdot)$, the proxy training data in (6) can be used to augment training for the image estimator $f(\cdot)$, now with *full supervision* from the proxy images as:

$$\mathcal{L}_{\text{prox}:x} = \frac{1}{T} \sum_t \sum_{i=1}^2 \rho\left(f(y_{t:i}^+) \;-\; x_{t:i}^+\right). \tag{8}$$

This loss can be used even in the non-blind training setting, and provides a means of generating additional training data with more pairings of image and measurement parameters. Also note that although our proxy images $x_{t:i}^+$ are approximate estimates of the true images, they represent the ground-truth for the synthetically generated observations $y_{t:i}^+$. Hence, the losses $\mathcal{L}_{\text{prox}:\theta}$ and $\mathcal{L}_{\text{prox}:x}$ are approximate only in the sense that they are based on images that are not sampled from the true image distribution $p_x(\cdot)$. And the effect of this approximation diminishes as training progresses, and the image estimation network produces better image predictions (especially on the training set).

Our overall method randomly initializes the weights of the image and parameter networks $f(\cdot)$ and $g(\cdot)$, and then trains them with a weighted combination of all losses: $\mathcal{L}_{\text{swap}} + \gamma\mathcal{L}_{\text{self}} + \alpha\mathcal{L}_{\text{prox}:\theta} + \beta\mathcal{L}_{\text{prox}:x}$, where the scalar weights $\alpha, \beta, \gamma$ are hyper-parameters determined on a validation set. For non-blind training (of blind estimators), only the image estimator $f(\cdot)$ needs to be trained, and $\alpha$ can be set to 0.

Table 1: Performance (in PSNR dB) of various methods for compressive measurement reconstruction, on BSD68 and Set11 images for different compression ratios.

| Method | Supervised | BSD68 | | | Set11 | | |
|---|---|---|---|---|---|---|---|
| | | 1% | 4% | 10% | 1% | 4% | 10% |
| TVAL3 [15] | ✗ | - | - | - | 16.43 | 18.75 | 22.99 |
| BM3D-AMP [20] (patch-wise) | ✗ | - | - | - | 5.21 | 18.40 | 22.64 |
| BM3D-AMP [20] (full-image) | ✗ | - | - | - | 5.59 | 17.18 | 23.07 |
| ReconNet [12] | ✓ | - | 21.66 | 24.15 | 17.27 | 20.63 | 24.28 |
| ISTA-Net+ [30] | ✓ | 19.14 | 22.17 | 25.33 | 17.34 | 21.31 | 26.64 |
| Supervised Baseline (Ours) | ✓ | **19.74** | **22.94** | **25.57** | **17.88** | **22.61** | **26.74** |
| Unsupervised Training (Ours) | ✗ | 19.67 | 22.78 | 25.40 | 17.84 | 22.20 | 26.33 |
| Unsupervised Training (Ours) *ablation without self-loss* | ✗ | 19.59 | 22.73 | 25.32 | 17.80 | 22.10 | 26.16 |

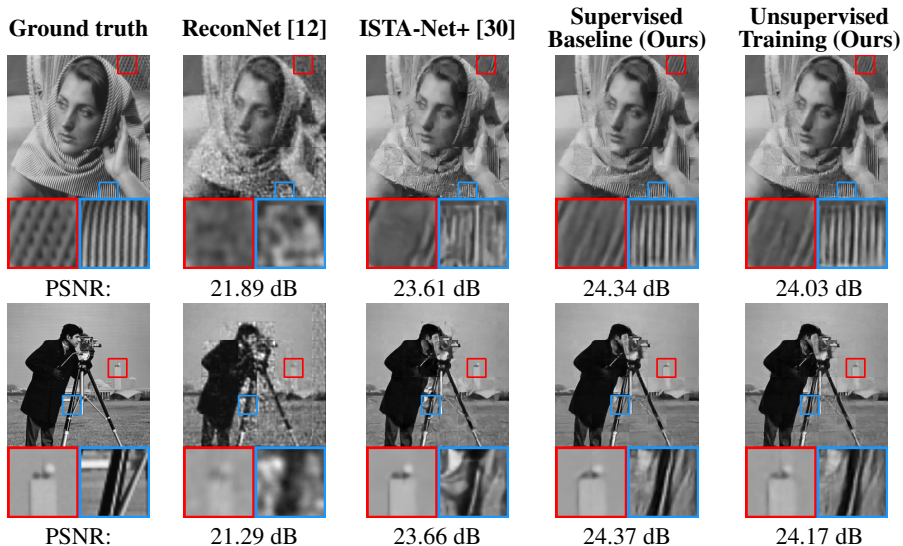

Figure 2: Images reconstructed by various methods from compressive measurements (at 10% ratio).

## 4 Experiments

We evaluate our framework on two well-established tasks: non-blind image reconstruction from compressive measurements, and blind deblurring of face images. These tasks were chosen since large training sets of ground-truth images *is* available in both cases, which allows us to demonstrate the effectiveness of our approach through comparisons to fully supervised baselines. The source code of our implementation is available at `https://projects.ayanc.org/unsupimg/`.

### 4.1 Reconstruction from Compressive Measurements

We consider the task of training a CNN to reconstruct images from compressive measurements. We follow the measurement model of [12, 30], where all non-overlapping $33 \times 33$ patches in an image are measured individually by the same low-dimensional orthonormal matrix. Like [12, 30], we train CNN models that operate on individual patches at a time, and assume ideal observations without noise (the supplementary includes additional results for noisy measurements). We train models for compression ratios of $1\%, 4\%$, and $10\%$ (using corresponding matrices provided by [12]).

We generate a training and validation set, of 100k and 256 images respectively, by taking $363 \times 363$ crops from images in the ImageNet database [26]. We use a CNN architecture that stacks two U-Nets [24], with a residual connection between the two (see supplementary). We begin by training our architecture with full supervision, using *all overlapping* patches from the training images, and an $L_2$ loss between the network's predictions and the ground-truth image patches. For unsupervised training

with our approach, we create two partitions of the original image, each containing *non-overlapping* patches. The partitions themselves overlap, with patches in one partition being shifted from those in the other (see supplementary). We measure patches in both partitions with the same measurement matrix, to yield two sets of measurements. These provide the diversity required by our method as each pixel is measured with a different patch in the two partitions. Moreover, this measurement scheme can be simply implemented in practice by camera translation. The shifts for each image are randomly selected, but kept fixed throughout training. Since the network operates independently on patches, it can be used on measurements from both partitions. To compute the swap-measurement loss, we take the network's individual patch predictions from one partition, arrange them to form the image, and extract and then apply the measurement matrix to shifted patches corresponding to the other partition. The weight $\gamma$ for the self-measurement loss is set to 0.05 based on the validation set.

In Table 1, we report results for existing compressive sensing methods that use supervised training [12, 30], as well as two methods that do not require any training [15, 20]. We report numbers for these methods from the evaluation in [30] that, like us, reconstruct each patch in an image individually. We also report results for the algorithm in [20] by running it on entire images (i.e., using the entire image for regularization while still using the per-patch measurement measurement model). Note that [20] is a D-AMP-based estimator (and while slower, performs similarly to the learned D-AMP estimators proposed in [21, 33] as per their own evaluation).

Evaluating our fully supervised baseline against these methods, we find that it achieves state-of-the-art performance. We then report results for training with our unsupervised framework, and find that this leads to accurate models that only lag our supervised baseline by 0.4 db or less in terms of average PSNR on both test sets—and in most cases, actually outperforms previous methods. This is despite the fact that these models have been trained without any access to ground-truth images. In addition to our full unsupervised method with both the self- and swap- losses, Table 1 also contains an ablation without using the self-loss, which is found to lead to a slight drop in performance. Figure 2 provides example reconstructions for some images, and we find that results from our unsupervised method are extremely close in visual quality to those of the baseline model trained with full supervision.

## 4.2    Blind Face Image Deblurring

We next consider the problem of blind motion deblurring of face images. Like [27], we consider the problem of restoring $128 \times 128$ aligned and cropped face images that have been affected by motion blur, through convolution with motion blur kernels of size upto $27 \times 27$, and Gaussian noise with standard deviation of two gray levels. We use all 160k images in the CelebA training set [17] and 1.8k images from Helen training set [13] to construct our training set, and 2k images from CelebA val and 200 from the Helen training set for our validation set. We use a set of 18k and 2k random motion kernels for training and validation respectively, generated using the method described in [4]. We evaluate our method on the official blurred test images provided by [27] (derived from the CelebA and Helen test sets). Note that unlike [27], we do not use any semantic labels for training.

In this case, we use a single U-Net architecture to map blurry observations to sharp images. We again train a model for this architecture with full supervision, generating blurry-sharp training pairs on the fly by pairing random of blur kernels from training set with the sharp images. Then, for unsupervised training with our approach, we choose two kernels for each training image to form a training set of measurement pairs, that are kept fixed (including the added Gaussian noise) across all epochs of training. We first consider non-blind training, using the true blur kernels to compute the swap- and self-measurement losses. Here, we consider training with and without the proxy loss $\mathcal{L}_{\text{prox}:x}$ for the network. Then, we consider the blind training case where we also learn an estimator for blur kernels, and use its predictions to compute the measurement losses. Instead of training a entirely separate network, we share the initial layers with the image UNet, and form a separate decoder path going from the bottleneck to the blur kernel. The weights $\alpha, \beta, \gamma$ are all set to one in this case.

We report results for all versions of our method in Table 2, and compare it to [27], as well as a traditional deblurring method that is not trained on face images [28]. We find that with full supervision, our architecture achieves state-of-the-art performance. Then with non-blind training, we find that our method is able to come close to supervised performance when using the proxy loss, but does worse without—highlighting its utility even in the non-blind setting. Finally, we note that models derived using blind-training with our approach are also able to produce results nearly as accurate as those trained with full supervision—despite lacking access both to ground truth image data, and knowledge

Table 2: Performance of various methods on blind face deblurring on test images from [27].

| Method | Supervised | Helen | | CelebA | |
|---|---|---|---|---|---|
| | | PSNR | SSIM | PSNR | SSIM |
| Xu *et al.* [28] | ✗ | 20.11 | 0.711 | 18.93 | 0.685 |
| Shen *et al.* [27] | ✓ | 25.99 | 0.871 | 25.05 | 0.879 |
| Supervised Baseline (Ours) | ✓ | **26.13** | **0.886** | **25.20** | **0.892** |
| Unsupervised Non-blind (Ours) | ✗ | 25.95 | 0.878 | 25.09 | 0.885 |
| Unsupervised Non-blind (Ours) *without proxy loss* | ✗ | 25.47 | 0.867 | 24.64 | 0.873 |
| Unsupervised Blind (Ours) | ✗ | 25.93 | 0.876 | 25.06 | 0.883 |

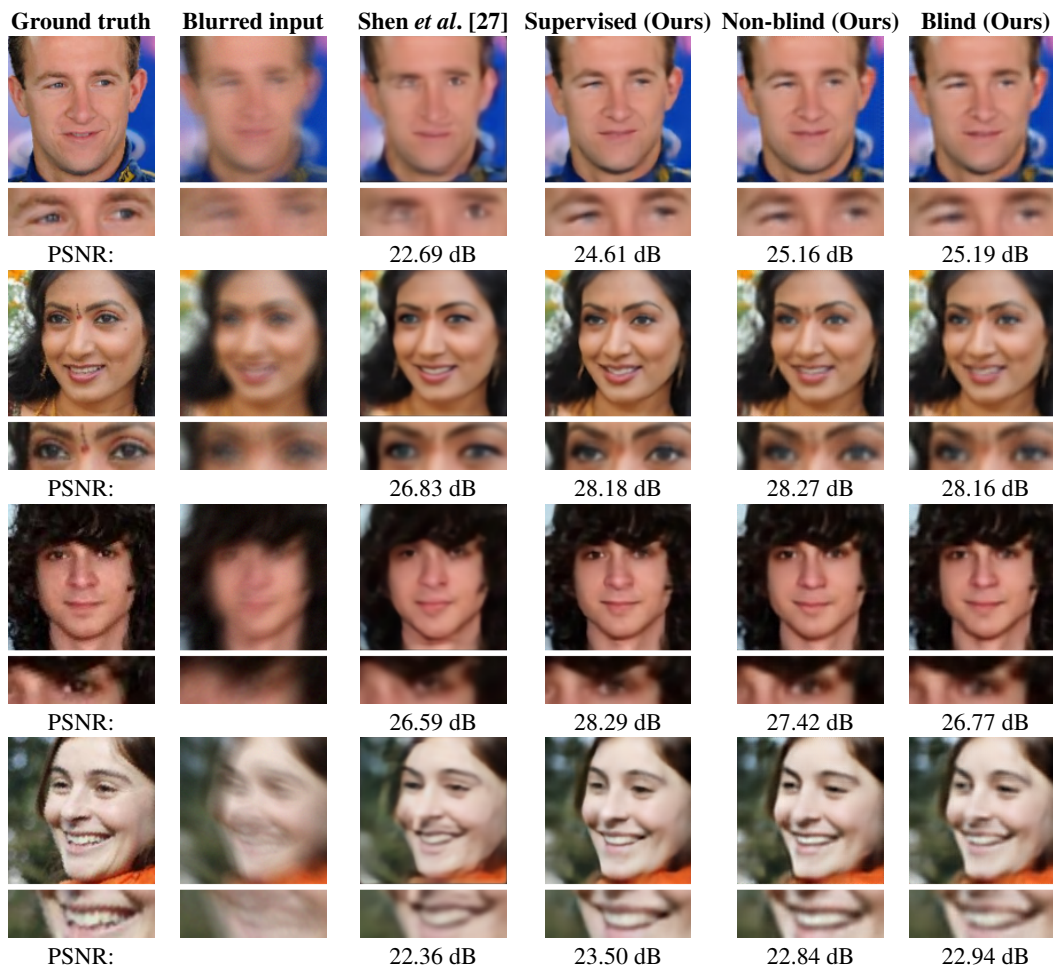

Figure 3: Blind face deblurring results using various methods. Results from our unsupervised approach, with both non-blind and blind training, nearly match the quality of the supervised baseline.

of the blur kernels in their training measurements. Figure 3 illustrates this performance qualitatively, with example deblurred results from various models on the official test images. We also visualize the blur kernel estimator learned during blind training with our approach in Fig. 4 on images from our validation set. Additional results, including those on real images, are included in the supplementary.

| Ground Truth | Blurred | Predictions | Ground Truth | Blurred | Predictions |

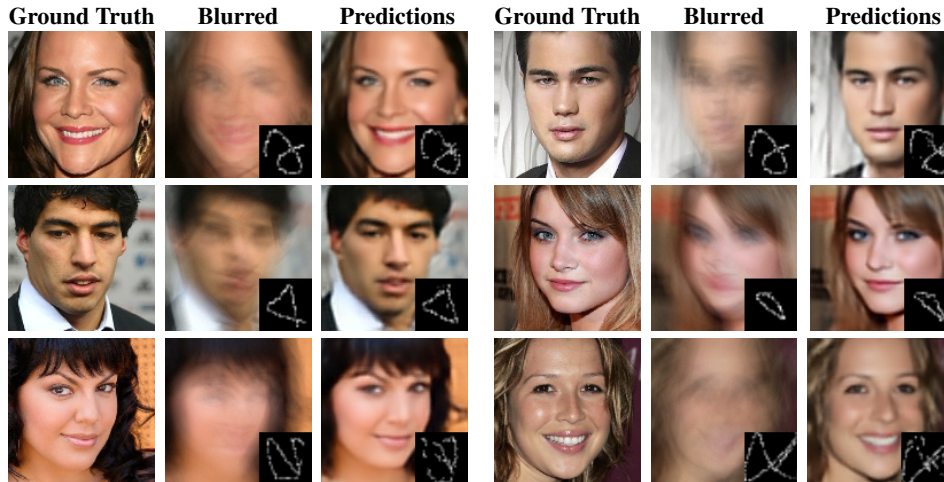

Figure 4: Image and kernel predictions on validation images. We show outputs of our model's kernel estimator, that is learned as part of blind training to compute swap- and self-measurement losses.

## 5  Conclusion

We presented an unsupervised method to train image estimation networks from only measurements pairs, without access to ground-truth images, and in blind settings, without knowledge of measurement parameters. In this paper, we validated this approach on well-established tasks where sufficient ground-truth data (for natural and face images) was available, since it allowed us to compare to training with full-supervision and study the performance gap between the supervised and unsupervised settings. But we believe that our method's real utility will be in opening up the use of CNNs for image estimation to new domains—such as medical imaging, applications in astronomy, etc.—where such use has been so far infeasible due to the difficulty of collecting large ground-truth datasets.

**Acknowledgments.** This work was supported by the NSF under award no. IIS-1820693.

## Footnotes

[1]Note that at test time, the trained network only requires one observation as input as usual.

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
