[Supplementary Material]

**Supplementary Material: Training Image Estimators without Image Ground-Truth**

*Zhihao Xia and Ayan Chakrabarti, Washington University in St. Louis*
`{zhihao.xia,ayan}@wustl.edu`

# A    Additional Results

## A.1    Reconstruction from Compressive Measurements

We include additional results for the case where the compressive measurements are corrupted by additive white Gaussian noise. When training with full supervision, we generate the noisy measurement on the fly resulting in many noisy compressed measurements for each image. But for unsupervised training with our approach, we keep the noise values (along with the measurement parameters) for each image fixed across all training epochs. We show results for Gaussian noise with different standard deviations in Table 3 for the 10% compression ratio. Again, our unsupervised training approach comes close to matching the accuracy of the fully supervised baseline. Figure 5 shows example reconstructions for this case.

Table 3: Performance (in PSNR dB) of our supervised baseline and proposed unsupervised method for noisy compressive measurement reconstruction, on BSD68 and Set11 images for different noise levels and compression ratio 10%.

| Method | BSD68 | | | | Set11 | | | |
|---|---|---|---|---|---|---|---|---|
| | $\sigma_\epsilon$=0 | $\sigma_\epsilon$=0.1 | $\sigma_\epsilon$=0.2 | $\sigma_\epsilon$=0.3 | $\sigma_\epsilon$=0 | $\sigma_\epsilon$=0.1 | $\sigma_\epsilon$=0.2 | $\sigma_\epsilon$=0.3 |
| Supervised Baseline | 25.57 | 24.60 | 23.49 | 22.57 | 26.74 | 25.24 | 23.67 | 22.30 |
| Unsupervised Training | 25.40 | 24.41 | 23.12 | 21.99 | 26.33 | 24.94 | 23.21 | 21.79 |

## A.2    Blind Face Image Deblurring

We begin by using visualizations of the Fourier magnitude spectra of blur kernels in deblurring to provide further intuition about the source of supervision in our method. In Fig. 6, we consider two example kernels used to obtain measurements during training. We see that their magnitude spectra have "zeros" (i.e., values that are equal or very close to 0)—which implies that their corresponding linear measurement parameters $\theta$ are low-rank and thus non-invertible. But note that their zeros are at different frequency components. We also see that pairs of measurements by these kernels are also non-invertible because their average spectrum (i.e., the average of their individual magnitude spectra) also has zeros—since even though the kernels individually have zeros at different frequencies, a pair of kernels can still have some common frequencies where both have zero response. However, the average magnitude spectrum of kernels across the entire training set is more homogeneous and has no zeros—indicating that every frequency component is well observed by at least some reasonable fraction of kernels in the training set. Consequently, the matrix $Q = \sum \theta^T \theta$ is full-rank, and the swap-loss is able to provide complete supervision during training.

Next, we show additional face deblurring results from [3]'s test set in Fig. 7. Moreover, [3] also provides a dataset of real blurred images that are aligned, cropped, and scaled. While there is no ground-truth image data available for this set, we include example results from it in Fig. 8 for qualitative evaluation. We again find that results from models trained using our unsupervised approach are close in visual quality to those from our supervised baseline.

# B    Network Architectures and Details of Training

Both of our compressive measurement reconstruction and face deblurring networks are based on U-Net [2], featuring encoder-decoder architectures with skip connections. We use convolutional layers with stride larger than 1 for downsampling, and transpose convolutional layers for upsampling. Except for the last layer of each network, all layers are followed by batch normalization and ReLU. We use $L_2$ distance as $\rho(\cdot)$ for all losses for compressive measurement reconstruction, and the $L_1$ distance (again, for all losses) in blind face deblurring. All networks are trained with Adam [1]

Figure 5: Example reconstructions from noisy compressive measurements, with supervised and unsupervised models.

| Kernel 1 | Spectrum 1 | Kernel 2 | Spectrum 2 | Avg Spectrum of Kernels 1 & 2 | Avg Spectrum of all Kernels |

Figure 6: Visualization of Fourier magnitude spectrum of individual kernels, pairs of kernels, and the average over the entire training set. The Fourier magnitude spectrum of a blur kernel represents the inner-product $\theta^T\theta$ of the corresponding measurement parameter, with zeros or very low values in the spectrum indicating that the measurement parameter induced by the kernel is low rank. Here, we see that individual kernels, and even pairs of kernels, have zeros in their individual and average spectra and are low-rank. But the average spectrum across the full training set (equivalent to $Q$) is full rank.

Table 4: Detailed architecture of the U-Net used for compressive measurement reconstruction. We stack two such networks together, and the final image estimate is the sum of their outputs. All "upconv" layers correspond to transpose convolution, $\oplus$ implies concatenation, and unless indicated with "VALID", all layers use "SAME" padding.

| Input | Output | Kernel Size | # input channels | # output channels | Stride | Output Size |
|---|---|---|---|---|---|---|
| $\theta^T y$ or $\theta^T y \oplus$ U-Net-1 out | conv1 | 2 | 1 or 2 | 32 | 1 | 32 (VALID) |
| conv1 | conv2 | 4 | 32 | 64 | 2 | 16 |
| conv2 | conv3 | 4 | 64 | 128 | 2 | 8 |
| conv3 | conv4 | 4 | 128 | 256 | 2 | 4 |
| conv4 | conv5 | 4 | 256 | 256 | 2 | 2 |
| conv5 | conv6 | 4 | 256 | 256 | 2 | 1 |
| conv6 | upconv1 | 4 | 256 | 256 | 1/2 | 2 |
| conv5 $\oplus$ upconv1 | upconv2 | 4 | 512 | 256 | 1/2 | 4 |
| conv4 $\oplus$ upconv2 | upconv3 | 4 | 512 | 128 | 1/2 | 8 |
| conv3 $\oplus$ upconv3 | upconv4 | 4 | 256 | 64 | 1/2 | 16 |
| conv2 $\oplus$ upconv4 | upconv5 | 4 | 128 | 32 | 1/2 | 32 |
| conv1 $\oplus$ upconv5 | upconv6 | 2 | 64 | 32 | 1 | 33 (VALID) |
| upconv6 | end1 | 3 | 32 | 32 | 1 | 33 |
| end1 | end2 | 1 | 32 | 1 | 1 | 33 |

optimizer and a learning rate of $10^{-3}$. We drop the learning rate twice by $\sqrt{10}$ when the loss on the validation set flattens out. Training takes about one to two days on a 1080 Ti GPU.

**Compressive Reconstruction.** Our compressive measurement reconstruction network is a stack of two U-Nets, with the detailed configuration of each U-Net shown in Table 4. Given a compressed vector $y$ for a single patch and the sensing matrix $\theta$, we first compute $\theta^T y$ and reshape it to the original size of the patch (i.e., $33 \times 33$) and input this to the first U-Net. The second U-Net then takes as input the concatenation of $\theta^T y$ and the output from the first U-Net. Finally, we add the outputs of these two U-Nets to derive the final estimate of the image.

Our approach to deriving measurement pairs during training is visualized in Fig. 9.

**Face deblurring.** Our face deblurring network is also a U-Net that maps the blurred observation to a sharp image estimate of the same size. For blind training, we have an auxiliary decoder path to produce the kernel estimate (i.e., to act as $g(\cdot)$). The kernel decoder path has the same number of transpose convolution layers, but only the first few upsample by two and have skip connections, since the kernel is smaller. The remaining transpose convolution layers have stride 1, but increase spatial size (as they represent transpose of a 'VALID' convolution). The final output of the kernel decoder path is passed through a "softmax" that is normalized across spatial locations. This yields a kernel

| Ground truth | Blurred input | Shen *et al*. [3] | Supervised (Ours) | Non-blind (Ours) | Blind (Ours) |
|---|---|---|---|---|---|
| PSNR: | | 25.28 dB | 27.01 dB | 26.35 dB | 26.21 dB |
| PSNR: | | 22.06 dB | 23.76 dB | 23.77 dB | 23.96 dB |
| PSNR: | | 24.34 dB | 26.38 dB | 26.06 dB | 25.88 dB |
| PSNR: | | 24.84 dB | 26.20 dB | 27.05 dB | 26.77 dB |
| PSNR: | | 25.87 dB | 27.14 dB | 26.87 dB | 26.67 dB |
| PSNR: | | 29.04 dB | 30.10 dB | 29.61 dB | 29.05 dB |
| PSNR: | | 27.59 dB | 29.82 dB | 30.62 dB | 30.66 dB |

Figure 7: Additional face deblurring results from the test set from [3].

| Blurred input | Shen *et al.* [3] | Supervised (Ours) | Non-blind (Ours) | Blind (Ours) |

Figure 8: Face deblurring results on real blurry face images as provided by [3].

Figure 9: Forming pairs of compressive mesaurements with shifted partitions. We form our measurements by dividing the image into two shifted sets of overlapping patches, where the shifts for are sampled randomly for each training image, but kept fixed through all epochs of training. All patches, in both partitions, are measured with a common measurement matrix. This provides the required diversity of our method since each pixel in the image (except those near boundaries) are measured twice, differently within two different overlapping patches.

with elements that sum to 1 (which matches the constraint that the blur kernel doesn't change the average intensity, or DC value, of the image). The detailed architecture is presented in Table 5.

Table 5: Architecture of the U-Net used for blind face deblurring. The second decoder path that produces a kernel estimate (koutput) is only used for blind training.

| Input | Output | Kernel Size | # input channels | # output channels | Stride | Output Size |
|---|---|---|---|---|---|---|
| RGB | conv1 | 4 | 3 | 64 | 2 | 64 |
| conv1 | conv2 | 4 | 64 | 128 | 2 | 32 |
| conv2 | conv3 | 4 | 128 | 256 | 2 | 16 |
| conv3 | conv4 | 4 | 256 | 512 | 2 | 8 |
| conv4 | conv5 | 4 | 512 | 512 | 2 | 4 |
| conv5 | conv6 | 4 | 512 | 512 | 2 | 2 |
| conv6 | conv7 | 4 | 512 | 512 | 2 | 1 |
| conv7 | upconv1 | 4 | 512 | 512 | 1/2 | 2 |
| conv6 ⊕ upconv1 | upconv2 | 4 | 1024 | 512 | 1/2 | 4 |
| conv5 ⊕ upconv2 | upconv3 | 4 | 1024 | 512 | 1/2 | 8 |
| conv4 ⊕ upconv3 | upconv4 | 4 | 1024 | 256 | 1/2 | 16 |
| conv3 ⊕ upconv4 | upconv5 | 4 | 512 | 128 | 1/2 | 32 |
| conv2 ⊕ upconv5 | upconv6 | 4 | 256 | 64 | 1/2 | 64 |
| conv1 ⊕ upconv6 | output | 4 | 128 | 3 | 1/2 | 128 |
| conv7 | kupconv1 | 4 | 512 | 512 | 1/2 | 2 |
| conv6 ⊕ kupconv1 | kupconv2 | 4 | 1024 | 512 | 1/2 | 4 |
| conv5 ⊕ kupconv2 | kupconv3 | 4 | 1024 | 512 | 1/2 | 8 |
| conv4 ⊕ kupconv3 | kupconv4 | 4 | 1024 | 256 | 1/2 | 16 |
| conv3 ⊕ kupconv4 | kupconv5 | 4 | 512 | 128 | 1 | 19 (VALID) |
| kupconv5 | kupconv6 | 4 | 128 | 64 | 1 | 22(VALID) |
| kupconv6 | kupconv7 | 4 | 128 | 64 | 1 | 25(VALID) |
| kupconv7 | koutput | 3 | 128 | 64 | 1 | 27(VALID) |