[Reviews · NeurIPS 2019]

Reviewer 1



Originality: The paper is mainly based on the idea presented in [14] and could be considered a generalization of it. Section 3.2 is the part which makes this paper's originality clear. Quality: Quality is the issue which makes the reviewer to believe this paper is not ready for publication yet. Here are the issues: - First of all, there are few previous works on the exact same problem that are neither cited nor compared against in this manuscript. These papers even do not need either ground truth data or two sets of measurements (unlike the submitted paper) and have shown impressive results. Examples are but not limited to: -- Simultaneous compressive image recovery and deep denoiser learning from undersampled measurements by Zussip et al. -- Unsupervised Learning with Stein’s Unbiased Risk Estimator by Metzler et al. - Another major problem with this manuscript is that all the methods presented in the experiments section are patch-based methods and are not state-of-the-art methods in learning-based image recovery. Instead, there are other approaches that can recover images in whole (instead of patch-based recovery) and have better performance compared to ISTA-Net and ReconNet. However, authors have not compared their work against any of the non-patch-based methods. Please note that one problem which makes the applicability of patch-based methods limited is that they could not be applied in certain applications such as medical imaging. In other words, based on the structure of sensing matrices (e.g. Fourier matrices), sometimes we are not able to sense and reconstruct images patch by patch. Authors should at least compare against non-patch-based CS recovery methods even if their method cannot be edited in a way that works for non-patch-based applications. - If you take a look at Fig 8 in the supplementary, there is an overlap between two partitions during the training phase. Now if authors claim the compression ratio of 10%, because of the overlap between partitions, the effective compression ratio of common area between two partitions is more than 10% and that makes the comparison presented in Table 1 unfair. In general, if you use two sets of measurements: Y1 = AX + e1 Y2 = BX + e2 you can merge the two and have [Y1 Y2]^T = [A B]^T X + (e1+e2) Although the authors are not using the exact same X during the training phase for the two sets of measurements, these two partitions have a considerable overlapping and that makes the comparison with other patch-based methods problematic. Clarity: The paper is very well-written and well-organized. Significance: The problem that is studied by the authors is an important problem. However, there are issues with the experimental results that should be addressed to consider the proposed approach significant and the reviewer believes that this paper is not ready for publication yet. ------------------------------------------------------------ Post Rebuttal Comments: Thanks to the authors for clarifications in their rebuttal, specifically for providing the new experimental results in this short amount of time. It would be great if authors could add the points they mentioned in the rebuttal to the final version of their paper as well. My score is now updated.

Reviewer 2



Summary: The authors present an extension of Noise2Noise that is able to deal with (in addition to noise) parametric linear image degradations such as blur. As in Noise2Noise the authors require only pairs of coresponding noisy and degraded images. Here, the to images are assumed to be created from the same clean image by applyinng different parametric distortions (such as a convolution with two different blur kernels) and then adding different instanciations of zero centered noise. The authors distinguish bettween a non-blind and a blind setting. In the non-blind setting the parameters of the degradations are known. During training the authors use their network to process both images and then apply the corresponding other parametric distortion to each of the two processed images. The loss is then calculated as the squared error between the final results. In the blind setting the authors employ a second network to estimate the parameters of the degredation. It is trained in a supervised fashion on simulated training data, which is created from results of the first network obtained during the training process. The authors evaluate their method on two different datasets for the tasks of reconstruction from compressed measurement and face deblurring. The results are comparable to the supervised state-of-the-art baselines. Originality: + The manuscript presents an original and important extension of Noise2Noise. Clarity: + The paper is extremely clear in its presentation and well written. Significance: + I think the overall significance of the manuscript is quite high. The general idea of applying Noise2Noise in a setting witch includes (next to the noise) additional linear distortions of the image could open the door for new applications. Especially if the assumption of linearity were to be dropped in future research, such a method might be used to reduce difficult reconstruction artifacts in methods like tomography or super resolution microscopy. Quality: + I think the manuscript and the presented experiments are technically and theoretically sound. However, there are some open questions regarding the evaluation of the method: - I find it remarkable that the fully supervised baseline outperforms the other state-of-the-art methods in both experiments. How can this be explained? Is this due to the unconventional architecture (two concatenated u-nets) used by the authors? If this is the case, it seems like a separate additional contribution, which is not clearly stated. Final Recommendation: Considering that the paper introduces an original, highly significant extension to Noise2Noise and validates the approach via sufficient experiments, I recommend to accept the paper. ------------------------------------------ Post Rebuttal: I feel that my questions have been adequately addressed and I will stick with my initial rating.

Reviewer 3



In this paper, the authors introduce a method to train deep image estimators in an unsupervised fashion. The method is based on the same ideas as the recently introduced noise2noise method (Lehtinen et al, 2018) but is generalized to deal with a degradation model that produces linear noisy observations from the latent clean image that one wants to recover. The training avoids using ground-truth images by using two observations of the same latent image under different linear operators (i.e., compressive measurements or motion blur) and different noise realizations. The paper first introduces a non-blind training scheme where the linear operators are known. Then, for blind training the linear operators are estimated, using a second network (subnetwork), and used as a proxy to emulate the non-blind scheme. The whole system is trained by carefully avoiding updating all the parameters together (to avoid a vicious loop). Experimental evidence on two (synthetic) problems shows that the unsupervised (non-blind and blind) training scheme produces similar results as the fully supervised training. Training neural networks to reconstruct images without having access to ground truth data is a major challenge. Recent work (Soltanayev and Chun,2018; Lehtinen et al, 2018) have shown that this is possible in the particular setting of image denoising. This work generalizes the ideas behind Lehtinen et al, 2018 to cope with a linear model (where denoising can be seen as a particular case). This is an interesting paper that proposes several ideas to avoid the restrictions imposed in the Lehtinen et al. procedure. The paper is generally well-written, but there are a few sections that could be improved with more discussion. In fact, the major weakness of the paper is the lack of analysis. -- Swap Loss (Eq (3) and Eq (4)). The performance of the method strongly depends on the matrix Q, but there's no analysis on this. In particular, I can imagine that in the case where the linear operator is formed using random orthogonal matrices, Q will be close to the identity. But, what about other cases? For example, in the case of motion blur, the Q matrix will be related to the power spectrum of the motion kernel process (Q is the autocorrelation). Since it is essentially a random walk, I can imagine that it will have a linear decay with the frequency. This implies that the swap loss is not penalizing high-frequency errors so, the training scheme won't help to recover high-frequency details. The paper does not discuss any of this but claims that if the measurements are different/complementary enough then this matrix will be full rank. More analysis is needed regarding Q. For example, authors could plot the spectrum of this matrix in the given experiments. Also, this matrix Q is probably one major limitation of the method. This is not discussed in the paper. -- Self Loss (Eq (5)). To complement the swap loss, the authors introduce a "self loss" that enforces self-consistency. This is not analyzed and it is not clearly motivated. The justification is just that it produces better results. For instance, in the particular scenario where the linear operator is the identity (i.e., denoising), this loss will enforce that the network f is close to the identity so learns to do nothing (keep the noise!). I understand that this is a particular case of the more general model, but this particular case is covered within the proposed framework. I would like to have more information regarding this loss, and in general what is this loss doing. For instance, the paper could provide results when this self-loss is not used. -- Prox:\theta Loss (Eq (7)) The authors propose to cope with the blind case by learning the linear operator (degradation) using a convnet (a separate branch of the network). I find it quite surprising how well this works (according to the experiments). I would like to see more analysis regarding the estimation of the blur (or more generally, the linear operator), in particular, a comparison to other works doing this (e.g., motion blur estimation using deep learning). This is not mentioned much, but this is a very difficult problem in itself. -- Experiments Deblurring Face images. Why is the method trained for deblurring face images? Would it work if trained in a more complex distribution, e.g., natural images? Noise. The performance gap between supervised and unsupervised training seems to increase with the level of noise. Could you elaborate on this? Also, in the experiment regarding deblurring face images, the level of noise is 2/255, which is pretty low. Have you tried with higher noise levels? Loss. When deblurring face images, all adopted losses are L1. In this case, the mathematical analysis in Eq (4) doesn't hold. Could you comment on this? Also, why did you choose L1 for this case? Are results with L2 norm much worse? Compressive measurements. The shifted partitions are generated in a very particular way. (Figure 8 in supplementary material). Is this really important? How sensitive are the results to this pattern? This is related to the matrix Q. Other comments: It could be interesting to compare the proposed loss with the losses used in multi-image variational deblurring, for example, see: Zhang, H., Wipf, D. and Zhang, Y., 2013. Multi-image blind deblurring using a coupled adaptive sparse prior. In Proceedings of the IEEE Conference on Computer Vision and Pattern Recognition (pp. 1051-1058) -------- After rebuttal. I believe that the manuscript will improve with the changes that the authors have committed to do. Additionally, I would like the authors to clearly list the limitations of the current approach and briefly discuss them (see e.g., in rebuttal document, l42 to l44, estimation of \ theta, motion kernel estimation; ). This will allow defining clearer future research directions for those who are willing to pursue this line of work. Since the authors have carefully addressed most of the questions and comments I am therefore updating my score to 7. I would like to see this paper presented at this venue!

[Author Response · NeurIPS 2019]

**Reviewer 1: - Unfair Experiments:** There has been a misunderstanding! *We use only one measurement (one partition with no overlap for CS) at test time.* The two measurements (two overlapping partitions for CS) are used *only for training.* The underlying network is setup to estimate the image from a single measurement. During training, we apply the network separately to the two measurements to get two estimates of each training image, and use these estimates jointly to compute our losses. But for evaluation on the test set, our network is given only one measurement as input. Thus, *all methods are provided identical inputs (one measurement, no overlap)* that matches the noted ratios for CS.

**- Other Recent Methods:** We will cite & discuss these concurrent works (note they were only on arXiv until recently: the BASP workshop Feb'2019 for Metzler, and CVPR Jun'2019 for Zhussip). Both papers propose approaches based on SURE (i.e., very different from ours) for unsupervised training from single measurements. But they are specific to D-AMP estimation: they train denoiser networks for use in unrolled AMP iterations for CS recovery. Our method needs pairs of measurements, but is a more general framework that can be used to train generic neural network estimators.

**Comparisons**: We can't directly compare to Zhussip since there is no code available yet. Instead, we'll add comparisons to BM3D-AMP which was their main baseline. Moreover, BM3D-AMP was also the best performing CS method in Metzler's evaluation (see their Tbl.2: their proposed methods were faster but had lower avg PSNRs than BM3D-AMP).

Below, we show results of BM3D-AMP on Set11 with a "patch-wise" evaluation from the ISTANet paper, as well as a "full-image" evaluation suggested by R1 (calling BM3D-AMP on the full image with measurements from all patches as input). We see that, even with full image restoration, BM3D-AMP performs worse than our unsupervised method by 3.26 dB at 10% (gaps are even wider at lower rates, as also noted in the ISTANet paper). As reference, BM3D-AMP is better than Metzler, and worse than Zhussip by only 1-1.5 dB, in their own CS evaluations.

| Method | 1% | 4% | 10% |
|---|---|---|---|
| BM3D-AMP Patch-wise (from ISTANet Paper) | 5.21 dB | 18.40 dB | 22.64 dB |
| BM3D-AMP Full Image (our evaluation) | 5.59 dB | 17.18 dB | 23.07 dB |
| Our Method (unsupervised) | 17.84 dB | 22.20 dB | 26.33 dB |

**- Only Patch-based Restoration:** Nothing in our framework restricts it to patch-based networks: the entire second half of our evaluation (on deblurring) is on a full-image restoration network. For CS, we chose patch-based restoration only to follow the protocol of the recent CVPR'18 ISTANet paper (also, patch-wise is just a experimental choice: ISTANet and our CS network could be easily adapted to treating the whole image as a large "patch"). For completeness, we will also add the full-image BM3D-AMP results above to our evaluation.

**Reviewer 2:** Thanks for your positive comments! Our main results are really the comparisons between the supervised and unsupervised versions of the same (our) network. In terms of architecture, we actually use concatenated U-Nets only for CS, inspired by ISTANet+ that also concatenated two networks. We had also tried a single U-Net for CS and got similar gaps between unsupervised and supervised (both did slightly worse than concatenated U-Nets). For deblurring, we actually have a single U-Net (with a second decoder path for kernel estimation used only in blind training). Also, rather than claim a contribution on a specific application (where the improvements from the architecture are relatively minor: 0.1-0.2dB for 10% CS and deblurring), we wanted to keep the focus on our general unsupervised framework.

**Reviewer 3:** Thanks for the detailed feedback! **- Q matrix / swap-loss:** We'll further discuss the full-rank requirement of $Q$ to give readers intuition (individual $\theta$ can be low-rank, but must be diverse in a way that $Q$ is full-rank). With motion-blur, the decay is along different orientations for different kernels. And as they are thin splines, most of the 'low-rankness' comes from a sinc-like pattern of zeros in frequency. Orientations and zero-frequencies vary across kernels, making $Q$ full-rank. We'll add images of individual kernel and average magnitude spectra to show this.

**- Self-loss:** The self-loss provides extra supervision per sample (although the swap loss is full-rank in expectation, it is low-rank per-sample). Note we use it in conjunction with the swap-loss, and any noise present will be independent in the two measurements. So for the identity case, the swap- and self-loss together will promote convergence to the mean of the two noisy measurements. We'll add an ablation to Table 1 w/o self-loss: performance drops by 0.04-0.17 dB.

**- prox:$\theta$ Loss:** Blind-training can indeed only be used when estimating $\theta$ is feasible. We'll clarify this assumption in the revision. For deblurring, kernel estimation is roughly as hard as deblurring (early deblurring methods first estimated the kernel & ran non-blind deconvolution), and is often reasonably successful (with supervised training).

**- Experiments:** *Face:* Class-specific deblurring (for faces, text, etc.) is a popular problem because it can perform better on that class compared to generic deblurring, by relying more on class-specific image priors (with supervised training). Since the CS experiments were already on general images, we felt this was an interesting second evaluation for our unsupervised method. *Noise:* The gap between supervised and unsupervised indeed grows with more noise, as supervised benefits from always training on noise-free GT. The deblurring noise level was set by the benchmark. *L1 Loss:* We used L1 because it is a common choice in deblurring networks (gives sharper results). (4) doesn't hold as is, but expected loss is still only minimized by an ideal estimate. *CS Measure:* We just use two randomly shifted partitions: each dividing the image into non-overlapping patches (chosen as it could be realized practically by moving the sensor).

[Meta-Review · NeurIPS 2019]

This work introduces a new method to learn image restoration methods from only corrupted data sets. It is an exciting idea that could potentially open up new applications for deep learning methods in settings where it is not possible to obtain ground truth data. Three reviewers initially assessed the work as 5/9/6. Based on a strong author rebuttal all reviewers took part in a discussion and two reviewers revised their score upwards, for a final assessment of 6/9/7. Overall this paper contains an exciting idea and is likely to stimulate the NeurIPS community to further consider the setting of learning only from corrupted data.